# Acoustic-Sensing-Based Attribute-Driven Imbalanced Compensation for Anomalous Sound Detection without Machine Identity

**DOI:** 10.3390/s23218984

**Published:** 2023-11-05

**Authors:** Yifan Zhou, Yanhua Long, Haoran Wei

**Affiliations:** 1Key Innovation Group of Digital Humanities Resource and Research, Shanghai Normal University, Shanghai 200234, China; 2Unisound AI Technology Co., Ltd., Beijing 100089, China; 3Department of ECE, University of Texas at Dallas, Richardson, TX 75080, USA

**Keywords:** acoustic sensing, condition monitoring, anomalous sound detection, attribute classification, imbalanced compensation

## Abstract

Acoustic sensing provides crucial data for anomalous sound detection (ASD) in condition monitoring. However, building a robust acoustic-sensing-based ASD system is challenging due to the unsupervised nature of training data, which only contain normal sound samples. Recent discriminative models based on machine identity (ID) classification have shown excellent ASD performance by leveraging strong prior knowledge like machine ID. However, such strong priors are often unavailable in real-world applications, limiting these models. To address this, we propose utilizing the imbalanced and inconsistent attribute labels from acoustic sensors, such as machine running speed and microphone model, as weak priors to train an attribute classifier. We also introduce an imbalanced compensation strategy to handle extremely imbalanced categories and ensure model trainability. Furthermore, we propose a score fusion method to enhance anomaly detection robustness. The proposed algorithm was applied in our DCASE2023 Challenge Task 2 submission, ranking sixth internationally. By exploiting acoustic sensor data attributes as weak prior knowledge, our approach provides an effective framework for robust ASD when strong priors are absent.

## 1. Introduction

Acoustic-sensing-based anomalous sound detection (ASD) has become an increasingly important technique for predictive maintenance and condition monitoring in industrial environments, especially with the emergence of Industry 4.0. ASD aims to detect anomalous noises in acoustic signals that may indicate a fault or deterioration in mechanical equipment. When machinery begins to degrade, the vibration and sounds emitted often change subtly before failure occurs. By identifying these anomalous acoustic patterns, ASD systems can provide early warning of impending faults, enabling proactive maintenance to avoid catastrophic breakdowns. Traditional manual acoustic monitoring is labor-intensive and prone to human variability. The emergence of automated ASD systems addresses these limitations, reducing personnel costs and providing more consistent machine health assessment.

In most real-world scenarios, abnormal samples cannot be obtained by damaging the machine, and the complex engineering environment introduces much noise into the sound samples. The operating settings of different machines are also diverse. Therefore, the main challenge of the acoustic-sensing-based ASD task is to detect anomalous sounds when only normal sound samples are provided as training data [1,2,3].

In addition to acoustic sensing data, anomaly detection and fault diagnosis methods designed for other data types are also worth considering as references. In [4], the authors innovatively utilized event-based cameras for anomaly detection, collecting vibration signals of machines in a contactless manner and providing a new perspective for machine condition monitoring. Facing similar challenges of imbalanced data and dynamic operations, the authors in [5] combined a self-supervised anomaly detector based on a local outlier factor (LOF) and a deep Q-network (DQN) supervised reinforcement learner to classify interturn short-circuit, local demagnetization, and mixed faults. Additionally, in the context of small datasets in industrial settings, the authors in [6] optimized the friction-drilling process through model ensembling in order to work with no complete information. Feature engineering is also crucial for rotary machine monitoring. The authors in [7] proposed a novel feature extraction method called weighted multi-scale fluctuation-based dispersion entropy for detecting faults in planetary gearboxes. In [8], permutation entropy was integrated with a flexible analytical wavelet transform for bearing defect detection. These real-world practices in industrial scenarios provide valuable references for ASD work.

To drive the development of acoustic-sensing-based ASD technology, a sub-challenge (Task 2) of ‘Unsupervised Detection of Anomalous Sounds for Machine Condition Monitoring’ has been launched in the IEEE AASP Challenge on Detection and Classification of Acoustic Scenes and Events (DCASE) since 2020 [9]. In the previous work of the DCASE Challenge Task 2, the anomaly detection system AutoEncoder (AE) [9] based on generative model was widely used due to its simple design and efficient inference. In addition to acoustic sensing tasks, AEs have also been extensively utilized as unsupervised anomaly detectors in many other application domains [10,11]. However, AE-based anomaly detection relies on the assumption that anomalies are difficult to reconstruct. Given the inherent denoising characteristics of AE [12], enhancing the representation capacity of AE may inadvertently treat anomalies as noise, thus constraining the anomaly detection performance reliant on AE. In addition to AE, other generative models, such as IDNN [13], Efficient GAN [14], and Glow_Aff [15], detect anomalies by modeling the distribution of normal sounds and determining whether the sound under test is within the range of the normal sound distribution. However, due to the complexity of anomalous sounds, it is difficult to model a stable distribution for anomaly detection [16], which limits the generative model.

Therefore, in order to better model the characteristics of normal sounds, systems based on discriminative models [17] were designed and achieved excellent performance. These models utilize powerful deep learning feature extraction networks, such as ResNet [18], MobileNetV2 [19], and STgram [20], for self-supervised classification tasks related to machine ID. During inference, abnormal samples are exposed due to the difficulty in classification, allowing for effective anomaly detection. Undoubtedly, in previous challenges regarding DCASE Task 2 evaluations, the competitive anomaly detection systems were also based on utilization of machine ID classification. The algorithm’s success is attributed to the high-quality classification boundaries established by leveraging strong prior knowledge of machine ID. However, in practical applications, obtaining such high-quality prior knowledge of machine ID is often unfeasible. This raises an important question: how can we adapt anomaly detection algorithms based on discriminative models to operate effectively under limited prior knowledge conditions?

Instead, we need to design anomaly detection algorithms under weak prior knowledge conditions. According to the task setting of DCASE2023 Task 2, we cannot obtain high-quality prior knowledge such as machine ID, but we can obtain the attributes information of each audio clip, such as microphone number, machine running speed, machine load status, etc. We define the attributes information of the audio clips as weak prior knowledge. Unfortunately, there is no free lunch in the world. Intuitively, these attribute labels are extremely imbalanced, complex categories, and cannot form clear classification boundaries, but they are more accessible in the real world.

In this paper, we propose the attribute-driven imbalanced compensation (AIC) method, aiming to overcome the disadvantages of weak prior knowledge and use attribute labels to build discriminative models for anomaly detection. Our main contributions are as follows: (1) we propose an attribute classifier using the weak prior knowledge, making the application of discriminative models possible when machine ID labels are limited. (2) We propose the imbalanced compensation strategy to solve the common problem of extreme sample imbalance in attribute labels. (3) We propose a score fusion method based on AIC to enhance the robustness of the model.

## 2. Proposed Method

The AIC framework we propose contains an imbalanced compensation module, *M* attribute classifiers, and an ensemble attribute anomaly detector. The overview of the overall framework is shown in Figure 1. The overall framework consists of training and testing stages. First, the raw data are augmented via imbalanced compensation separately for each attribute. In the training stage, a classifier is trained for the augmented data of each attribute with cross-entropy loss. In the testing stage, the augmented data are fed into the trained classifiers to obtain embeddings, with each attribute corresponding to one embedding space. Then, the embedding of the test samples is extracted through the trained classifier, and KNN is used to calculate the score in the ensemble attribute anomaly detector.

### 2.1. Attribute Classifier

Although the previous work using machine ID for classification achieved good results [1,2], it is often more limited in practical applications, such as only one machine is working. In this case, strong prior knowledge such as machine ID cannot be used. Nevertheless, machines still have weak prior knowledge that is easy to obtain, such as attributes in Table 1; both the ‘ToyCar’ and ‘ToyTrain’ have three types of weak attributes. Therefore, in this study, we propose to train the attribute classifier using such weak prior knowledge as attributes.

In real-world applications, different attributes of a machine may work under different operating status. These statuses can be easily collected and labeled. For example, as illustrated in Table 2, in the ToyCar dataset provided by the DCASE2023 Challenge, the attribute ‘Mic’ has two types of operating status: ‘1’ and ‘2’. These status labels can be used to train an anomaly classifier, allowing it to distinguish between different operating status for each attribute. This approach helps the anomaly classifier learn the details of the training dataset more comprehensively and deeply, similar to observing the same object from different perspectives. The operating status information provided by DCASE [21,22] naturally accompanies machine operation and is readily accessible.

Similar to previous machine ID classifiers [1], in our attribute classifier, we adopt the cross-entropy loss to classify each operating status of each attribute of each machine. As shown in Figure 1 training classifier step, and  both of Table 1 and Table 2, this can be described as training *M* classifiers for *M* attributes, where each classifier performs an $K_{m}$-class classification task and $K_{m}$ represents the total distinct types of operating status for the *m*-th attribute. We use ResNet18 [23] as the backbone encoder of the classifier to obtain the embedding of each attribute. The proposed *m*-th attribute classifier (AC) is trained with the cross-entropy loss function as
(1)$$L_{\mathrm{AC}}^{m}=-\frac{1}{NK_{m}}\sum_{i=1}^{N}\sum_{k_{m}=1}^{K_{m}}y_{ik_{m}}\log\left(p_{\theta_{E}}\left(x_{ik_{m}}\right)\right)$$
where *N* means the total input samples of the *m*-th attribute, $x_{ik_{m}}$, $y_{ik_{m}}$ represent the *i*-th input sample and its *k*-th operating status in attribute *m*. $p_{\theta_{E}}\left(\cdot \right)$ is the softmax output of the encoder with parameters $\theta_{E}$. As depicted in Figure 1 training detector step, following the independent training of the *M* attribute classifiers, we correspondingly learn *M* anomaly detectors, with each detector associated with one of the *M*-trained attribute classifiers. We utilize KNN as the anomaly detector, with the embeddings extracted by the trained classifiers used as the training data. During testing, the test sample is passed through each of the *M*-trained classifiers to obtain *M* embeddings, which are then fed into their corresponding *M* anomaly detectors to produce anomaly scores. The final aggregated anomaly score is obtained by taking the harmonic mean of the individual scores from each detector.

However, classifiers trained solely on operating status information for machine attributes often struggle to converge. Taking ToyCar dataset as an example, in Table 2, the information of training samples is expressed in the form of ‘Category: #samples’, and we can see that different attributes have varying numbers of operating status categories. For example, ‘Car model’ has 10 categories from A1 to E2, ‘Speed’ has 5 categories, controlled by voltage levels from 2.8 V to 4.0 V, and ‘Mic’ includes 2 categories, 1 and 2. Additionally, the number of samples for each operating status category is highly unbalanced. These factors make machine attributes weaker prior knowledge compared to machine IDs and pose several challenges with using operating status alone for training attribute classifiers: (1) the inconsistency in the number of attributes and operating status categories across different types of machines makes it difficult to establish consistent classification boundaries; (2) the severe sample imbalance within the same attribute and operating status category affects the classifier’s ability to accurately characterize normal samples; (3) during testing, the machine attributes and their corresponding operating status are unknown, further complicating the classification process. As a result, classifiers trained only on operating status are prone to misclassifying normal unseen samples as anomalies. To address this issue, we need to propose a method that strengthens the weak attribute knowledge as prior information in anomaly detection models.

### 2.2. Imbalanced Compensation

To solve the problem of severe unbalanced samples among different operating status shown in Table 2, in this section, we propose an imbalanced compensation module to enhance the proposed attribute classifier training. The module mainly includes two parts: (1) maximum expansion uniform sampling, and (2) robust data transformation. Figure 2 illustrates the schematic diagram of the effects of imbalanced compensation on acoustic-sensing-based ASD training data.

In Figure 2, different symbol shapes denote different categories with originally imbalanced number of samples. After maximum expansion and uniform sampling, the categories are balanced in terms of sample counts. Boundary shape changes following robust data transformation signify altered data distributions. In the first step, we identify the category with the maximum number of samples and expand all other categories to this level via oversampling. While this balances the quantities, the original data distributions remain unchanged, impeding effective training of the attribute classifier. Therefore, we subsequently apply robust data transformations, randomly augmenting the balanced data with 4 different techniques to alter the data distribution and simulate varied recording conditions. This enables successful training of the attribute classifier and enhances model robustness. In summary, our proposed pipeline tackles data imbalance through expansion and synthesizes robustness via data transformation, enabling learning from skewed real-world data.

The detail algorithm of our proposed imbalanced compensation module is presented in Algorithm 1. Given an unbalanced dataset with all *N* samples for one attribute of one machine, ${\left\{x_{ik_{m}}\right\}}_{i=1,k_{m}=1}^{N,K_{m}}$, which has $N_{k_{m}}$ samples, it is classified into the $k_{m}$-th category of operating status for each attribute, satisfying $N=\sum_{k_{m}=1}^{K_{m}}N_{k_{m}}$. With the $k_{m}$ varying, $N_{k_{m}}$ takes different values, resulting in an imbalance of data in each attribute.
**Algorithm** **1** Proposed imbalanced compensation method in *m*-th attribute**Input:** An unbalanced dataset of all *N* samples in *m*-th attribute, ${\left\{x_{ik_{m}}\right\}}_{i=1,k_{M}=1}^{N,K_{m}}$**Output:** A balanced dataset of *R* samples after imbalanced compensation (IC) in *m*-th attribute, ${\left\{{x_{ik_{m}}}^{IC}\right\}}_{i=1,k=1}^{R,K_{m}}$  1. Find the maximum count of operating status $T=\max N_{k_{m}}$  2: Calculate the sample increment $\Delta_{k_{m}}=T-N_{k_{m}}$ for each operating status  3: Expand sample numbers to $N^{*}=K_{m}\times T$  4: Get a balanced dataset ${\left\{{x_{ik_{m}}}^{MEUS}\right\}}_{i=1,k_{m}=1}^{N^{*},K_{m}}$ after maximum expansion uniform sampling (MEUS)  5: Sample *R* times in $\left\{{x_{ik_{m}}}^{MEUS}\right\}$  6: **for** i,k,m in $R,K_{m},M$ **do**  7:    ${x_{ik_{m}}}^{IC}=T_{4}\left(T_{3}\left(T_{2}\left(T_{1}\left({x_{ik_{m}}}^{MEUS}\right)\right)\right)\right)$  8: **end for**  9: Obtain the final dataset ${\left\{{x_{ik_{m}}}^{IC}\right\}}_{i=1,k_{m}=1}^{R,K_{m}}$ after imbalanced compensation

**Maximum Expansion Uniform Sampling**: As shown in Figure 2, we first introduce the maximum expansion uniform sampling to expand the original dataset to a balanced one. When we apply maximum expansion uniform sampling, we take the maximum value of $N_{k_{m}}$, set it to $T=\max N_{k_{m}}$. Then, for the $N_{k_{m}}$ samples in each operating status, we copy the data to increase the number of samples defined as $\Delta_{k_{m}}=T-N_{k_{m}}$. At this time, the total number of samples is expanded to $N^{*}=K_{m}\times T$, and the number of samples among each operating status reaches balance. The original samples $x_{ik_{m}}$ can be represented as ${x_{ik_{m}}}^{MEUS}$ after applying maximum expansion uniform sampling. Therefore, maximum expansion uniform sampling solves the problem of severe imbalance of training samples, allowing the classifier training to converge.

For example, the machine ToyCar has three attributes: ‘Car model’, ‘Speed’, and ‘Mic’. We train three separate classifiers for this machine. For the ‘Car model’ attribute, there are 10 categories ‘C1’–‘E1’ with extremely imbalanced quantities as shown in Table 2. With imbalanced compensation, we first apply maximum expansion uniform sampling. Specifically, we take the number of samples in the largest category ‘C1’, which is 215. Then, we resample each category to have 215 samples, making the number of samples balanced across categories. Similarly, we apply the same procedure to the other two attributes of ToyCar, balancing the number of samples for each category within every attribute.

**Robust Data Transformation**: Based on the enhanced maximum expansion uniform sampling, we then perform robust data transformation to augment the environment robustness of training data for improving the generalization ability of the resulting attribute classifier model. When we apply robust data transformation, we first sample *R* times after maximum expansion uniform sampling according to the law of large numbers. When *R* is large enough, the distribution of samples after *R* sampling is the same as the balanced sample distribution after maximum expansion uniform sampling. At the same time, each sampling is accompanied by 4 data augmentations, which are

**AddGaussianNoise**: Directly adding a noise signal obeying a zero-mean Gaussian distribution to the original audio signal in the time domain. In practical environments, many background noises can be regarded as additive noise. After such noise addition to the audio signal, it can capture various and complicate acoustic characteristics of real environments.**TimeStretch**: Changing the speed of audio without altering its pitch by a pre-defined rate. Here, we randomly applied rates in the range of [0.8, 1.25].**PitchShift**: Randomly increased or decreased the original pitch. Here, we vary the pitch by pre-defined semitones in the range of [−4, 4].**TimeShift**: Shifts the entire audio signal forward or backward. Here, the shift range was [−0.5, 0.5] of the total signal length.

The above augmentations are represented by $T_{1}$, $T_{2}$, $T_{3}$, and $T_{4}$, respectively. Specifically, these four data augmentations are applied to each sampled sample with a 50% probability, distorting the distribution after sampling. To some extent, robust data transformation simulates unknown samples and enhances the robustness of the classifier, making it less prone to errors when classifying completely unknown samples in the test set. In summary, the samples after robust data transformation can be expressed as ${x_{ik_{m}}}^{IC}=T_{4}\left(T_{3}\left(T_{2}\left(T_{1}\left({x_{ik_{m}}}^{MEUS}\right)\right)\right)\right)$.

Still taking ToyCar as an example, after applying maximum expansion uniform sampling, the number of samples is balanced across categories within each attribute. However, merely having a balanced quantity does not mean the data distribution is suitable for training classifiers. Therefore, we apply robust data transformation to transform the data. For each sample, there is a 50% chance to be applied with transformations of ‘AddGaussianNoise’, ‘TimeStretch’, ‘PitchShift’, and ‘TimeShift’, which can be combined. After applying robust data transformation to every sample, we discard the original samples. This completes imbalanced compensation.

In summary, the application of maximum expansion uniform sampling balanced the extremely imbalanced data between operating status. After maximum expansion uniform sampling, robust data transformation was applied, and each sampling was accompanied by 4 types of audio time domain conversion, which simulated various noises in real situations to some extent and improved the robustness of the model at the data level. In addition, the oversampling technique increased the sample size and achieved class balance, which solved the problem that the classifier was difficult to train.

### 2.3. Ensemble Attribute Anomaly Detector

Currently, in the field of anomaly detection, probability-based confidence methods [24,25] have been widely used. Specifically, this kind of method trains a classifier on normal samples for classification. During testing, normal samples will be classified into known categories by the classifier, while abnormal samples are difficult to distinguish. Intuitively, abnormal samples will receive a lower confidence score. However, incorrectly classifying samples during testing has a catastrophic impact on the performance of anomaly detection [26]. In addition, this method relies on the quality of model training, but, in practical use, the model is often overfitting, which will also affect the performance of anomaly detection.

Instead, we propose ensemble attribute anomaly detector. The key is to combine the traditional machine learning algorithm KNN with the classifier obtained from deep learning to improve the fault tolerance and robustness of the model for anomaly detection. The detailed algorithm of our proposed module is presented in Algorithm 2. Utilizing the data after imbalanced compensation, which are also used to train the classifiers, we train *M* separate KNN models. Specifically, the embeddings extracted from the trained *M* classifiers are leveraged as quality training data for each KNN. After training a KNN search tree for each model, test embeddings are extracted by passing the test sample through the corresponding classifier. The trained *m*-th KNN search tree is then utilized to find the topK nearest neighbors of the test embedding, constructing the set $T_{k}\left(e^{\mathrm{test}}\right)$. Subsequently, the Euclidean distance $d(e^{test})$ between $e^{test}$ and the samples in $T_{k}\left(e^{test}\right)$ is computed to obtain the distance matrix $D^{test}$. The anomaly score is calculated as the maximum value in the distance matrix
(2)$$S_{m}=\max\left(D^{test}\right)$$

**Algorithm** **2** KNN for anomaly detection from the perspective of the *m*-th attribute**Input:** Train data: ${\left\{{x_{ik_{m}}}^{IC}\right\}}_{i=1,k=1}^{R,K_{m}}$, Test data: $x^{test}$, Trained *m*-th classifier **Output:** Anomaly score $S_{m}$ for test sample $x^{test}$
  1: Extract embeddings ${\left\{{e_{ik_{m}}}^{IC}\right\}}_{i=1,k=1}^{R,K_{m}}$ by the *m*-th classifier from ${\left\{{x_{ik_{m}}}^{IC}\right\}}_{i=1,k=1}^{R,K_{m}}$
  2: Extract embedding $e^{test}$ by the *m*-th classifier from $x^{test}$
  3: Construct KNN search TREE from ${\left\{{e_{ik_{m}}}^{IC}\right\}}_{i=1,k=1}^{R,K_{m}}$
  4: Find topK nearest neighbors of $e^{test}$ using TREE
  5: Let $T_{k}\left(e^{test}\right)$ be the set comprising the topK nearest neighbors of $e^{test}$
  6: Compute distance $d(e^{test})$ between $x^{test}$ and samples in $T_{k}\left(e^{test}\right)$
  7: Obtain score set $D^{test}$ of test distance   8: **return** Anomaly score $S_{m}=\max\left(D^{test}\right)$


Finally, we take the mean of the results obtained for each attribute in the score domain to obtain the final ensemble anomaly value
(3)$$S=\frac{\sum_{m=1}^{M}S_{m}}{M}$$

Although KNN is a classic machine learning method, it is prone to curse of dimensionality when dealing with high-dimensional data, such as audio in this work. Naturally, we thought of using the outstanding feature extraction capability of deep neural networks to reduce the dimension of audio data to a low-dimensional space that KNN can characterize. Therefore, we use the attribute classifier mentioned above as a proxy task for the anomaly detection task and obtain supervision by distinguishing different operating status. After the training is completed, in the latent space, the samples of each operating status will gather together, while abnormal samples will be exposed because they are difficult to distinguish. Notably, our work trains multiple classifiers for each attribute, which enables each classifier to distinguish abnormal samples from different attribute perspectives. Such an operation improves the fault tolerance of anomaly detection. Even if one classifier makes a mistake, the results of other classifiers can compensate the errors.

Taking ToyCar as an example, after applying the imbalanced compensation module, we pretrain three separate classifiers for the three attributes, respectively. Meanwhile, using the training data after imbalanced compensation, three different embeddings are extracted via the three classifiers, which we term as embedding spaces. During testing, a test sample is fed into the three pretrained classifiers to obtain three test embeddings, each corresponding to one embedding space. Then, we apply a KNN algorithm to retrieve the topK nearest neighbors for each test embedding in its embedding space. The Euclidean distances between the test embedding and its topK neighbors are calculated. After obtaining the three sets of Euclidean distances, we take their average as the final anomaly score.

Therefore, for anomaly detection, the three different attributes provide three distinct detection perspectives for the same test sample. Fusing their scores allows the three perspectives to complement each other. Meanwhile, the imbalanced compensation technique enables classifier training and enhances classifier robustness through data augmentation. The resultant high-quality embeddings together with the proposed ensemble attribute anomaly detector boost anomaly detection performance.

## 3. Experimental Setup

### 3.1. Datasets

We evaluate our proposed approach on the development dataset provided for the DCASE2023 Challenge Task 2 [3], which contains two subsets: ToyADMOS2 [21] and MIMII DG [22]. The development dataset includes normal and anomalous operating sounds of seven types of machines recorded in single-channel, including Fan, Gearbox, Bearing, Slide rail (slider), Valve, ToyCar, and ToyTrain. For each of the 7 machine types, the dataset provides (1) 990 normal sound clips of 10 s length downsampled to 16 kHz for training in the source domain, (2) 10 normal sound clips for training in the target domain, and (3) 100 clips each of normal and anomalous sounds for testing. The source/target domain labels and attribute labels in the train dataset of each sample are provided, while the test dataset is not. The overview of datasets is shown in Figure 3.

This work focuses on attribute labels, which are mentioned in Section 2. Unlike machine IDs that may be unavailable, attribute labels are labels extracted from metadata. Attributes such as operating speed, operating voltage, etc., necessarily accompany the operation of the machine, so obtaining such labels is actually feasible. Unfortunately, there is no free lunch in the world, and such labels are extremely imbalanced and inconsistent, posing challenges to the design of our anomaly detection system. The various labels in the dataset are shown in Figure 4.

### 3.2. Evaluation Metrics

To evaluate the performance of the proposed model, we adopt the area under the curve (AUC) of the receiver operating characteristic (ROC) as the evaluation metric [9]. The ROC curve is created by plotting the true positive rate (TPR) against the false positive rate (FPR) at various threshold settings. The AUC measures the entire two-dimensional area underneath the ROC curve, which represents the degree or measure of separability between normal and anomalous instances. An AUC of 1 represents a perfect classifier, while an AUC of 0.5 represents a worthless classifier. Compared with metrics such as accuracy, AUC provides a more comprehensive evaluation of the model’s performance in the imbalance scenario. Therefore, AUC is more suitable for evaluating anomaly detection methods where negative samples dominate. Furthermore, the *p*AUC is also used in this work, which is calculated as the AUC over a low FPR range [0, *p*]. The AUC and *p*AUC are defined as
(4)$$\mathrm{AUC}=\frac{1}{N_{-}N_{+}}\sum_{i=1}^{N_{-}}\sum_{j=1}^{N_{+}}H\left(S\left(x_{j}^{+}\right)-S\left(x_{i}^{-}\right)\right)$$
(5)$$p\mathrm{AUC}=\frac{1}{\left\lfloor pN_{-}\right\rfloor N_{+}}\sum_{i=1}^{\left\lfloor pN_{-}\right\rfloor }\sum_{j=1}^{N_{+}}H\left(S\left(x_{j}^{+}\right)-S\left(x_{i}^{-}\right)\right)$$
where $\left\lfloor \cdot \right\rfloor$ is the flooring function, $S(x_{i}^{-})$ and $S(x_{j}^{+})$ mean the score of normal and anomalous test clips, respectively. $N_{-}$ and $N_{+}$ represent the number of them, respectively. Otherwise, the function H(x) is
(6)$$H\left(x\right)=\left\{\begin{array}{c} 1,x>0\\ 0,x\leq 0 \end{array}\right.$$

In practical acoustic-sensing-based ASD scenarios, a lower FPR is required. Therefore, we set p=0.1.

### 3.3. Implementation Details

For data preprocessing, we first converted the raw audio signals to log-mel-spectrogram using a short-time Fourier transform (STFT) with a window size of 1024 and a hop length of 512. A mel filterbank with 128 filters was applied and the magnitude of the STFT was converted to decibels. In this study, we use 128-dimensional log-mel-spectrogram as input features for the classifier. We adopt ResNet18 [23] as the backbone to design our classifier. The model is optimized using the Adam [27] optimizer with a learning rate of 0.0001 and trained for 15 epochs with a batch size of 128. In addition, the data augmentation used in the IC module uses the audiomendation [28] toolkit, and the KNN training uses the Pyod toolkit [29], the topK = 5. In this work, the number of samples *R* for the imbalanced compensation module is set to 4096.

## 4. Experimental Results

In this section, we present a comprehensive analysis of the experimental results. We conducted experiments on the development dataset released in DCASE2023 Challenge Task 2 and compared the experimental results with the official baseline AE. In addition, to verify the effectiveness of the imbalanced compensation module, we applied the module on the AE baseline, called AEIC, and obtained competitive performance. Finally, we ensemble AEIC with our AIC model, whose performance ranked sixth in DCASE2023 Challenge Task 2.

### 4.1. Signal Analysis

Considering the abstract nature of audio signals, we are unable to analyze them through direct observation of the waveforms. Furthermore, the audio in acoustic-sensing-based ASD tasks cannot be distinguished as normal or anomalous through the human ear. Therefore, we transform the signals into the frequency domain through STFT and observe them in the Mel scale, namely the mel-spectrogram.

As shown in the Figure 5, we plot the mel-spectrograms of the audio signals from seven machines. Based on the mel-spectrograms, we can simply classify the audio signals of these seven machines into stationary and non-stationary signals. It is noteworthy that the signal of the Slider machine appears to be non-stationary but is actually a periodic stationary signal [30]. The specific classification is as follows:**Stationary signals:** ToyCar, Bearing, Fan, Slider**Non-stationary signals:** ToyTrain, Gearbox, Valve

### 4.2. Results

The proposed AIC model, as illustrated in Figure 1, utilizes the weak attribute labels to train a classifier without machine ID and embeds the data with the classifier to expose anomalies. The experimental results are shown in Table 3, where AE-MSE and AE-MAHA are two official baselines. Both utilize autoencoders trained solely on normal samples with an MSE loss function. The difference lies in the testing phase, where AE-MSE uses the MSE as the anomaly score while AE-MAHA employs the Mahalanobis distance. AC is the result of direct attribute classification without the IC module.

For evaluation metrics, AUCs and AUCt denote the AUC of the model on the source domain data and target domain data, respectively. *p*AUC represents AUC at low FPR, as mentioned in Section 3. To measure the model performance under these three metrics of AUCs, AUCt, and *p*AUC, we take the harmonic mean of them, denoted by ‘hmean’. Similarly, to measure the model performance across the seven machines of data, we take the harmonic mean of each metric across the seven machines. This is also denoted by ‘hmean’ for consistency. By taking the harmonic mean of metrics on each subset, we summarize the performance across machines into a single representative value.

The experimental results show that the proposed AIC model is competitive compared with the two AE-based baselines, demonstrating the potential of discriminative models based on weak attribute labels for anomaly detection tasks. Notably, attribute classifier performing direct classification of the raw attribute labels achieved poor performance, indicating that the extremely imbalanced sample and other issues of the attribute such as weak labels make the classifier untrainable. However, the AIC model with the imbalanced compensation module, compared with attribute classifier, gained significant performance improvements. This shows that the proposed imbalanced compensation module effectively alleviated the weak label problem of the attributes. Moreover, as observed in the baseline results and other previous findings in the literature [32], it is interesting to find that the experimental results of the seven machines show inconsistent patterns. For example, the best model performance on ToyCar is not the same one as with ToyTrain. This is mainly because the seven machines have different acoustic characteristics, which makes it difficult to design a universally applicable model.

### 4.3. Imbalanced Compensation Module Analysis

As analyzed above, the proposed imbalanced compensation module plays a crucial role in the entire anomaly detection model. Therefore, this section further explores the effectiveness of the imbalanced compensation module and how to determine the number of samples *S* in the imbalanced compensation module.

First, the imbalanced compensation module is applied to two official baselines, AE-MSE and AE-MAHA. Although AE itself does not utilize attribute labels, it has source domain and target domain labels, with imbalanced scenarios similar to those of attribute labels. Therefore, the imbalanced compensation module is applied according to domain labels. As shown in Table 4, AEIC-MSE and AEIC-MAHA are two baselines with the applied imbalanced compensation module, which achieved significant improvements over the original baselines. This demonstrates the universal effectiveness of the proposed imbalanced compensation module for both generative and discriminative models.

Furthermore, the performance changes in the AIC model with different numbers of imbalanced compensation samples *R* 1024, 2048, 4096, 8192 are explored. The harmonic mean of AUC on seven machines is used as the evaluation metric. We finally chose 4096 samples. As shown in Figure 6, surprisingly, the model performance of four stable signal machines, ToyCar, Bearing, Fan, and Slider, increases with the increasing number of samples. In contrast, the performance of three non-stable signal machines, ToyTrain, Gearbox, and Valve, cannot be improved with more samples. This could be because the time-domain data transformation of non-stable signals causes distortion, while that of stable signals helps to improve the model’s robustness.

### 4.4. Visualization

To better demonstrate the performance of AIC, t-SNE [33] is used to visualize the training set and test set. As shown in Figure 7, dots of different colors represent normal samples of the source domain in the training set, normal samples of the target domain in the training set, normal samples of the source domain in the test set, normal samples of the target domain in the test set, abnormal samples of the source domain in the test set, and abnormal samples of the target domain in the test set, respectively. The embeddings of attribute classifier and AIC are extracted separately for visualization. Since the model training objective is attribute classification, abnormal samples are hard to be classified into any category, thereby exposing the abnormal samples. This is manifested as abnormal samples are far from normal samples, forming lower-density areas in the visualization.

Comparing Figure 7a,b, taking Fan as an example, it is found that, for attribute classifier without the imbalanced compensation module, a small portion of normal samples are misclassified into areas close to abnormal samples, which will damage the anomaly detection performance [26]. However, for AIC with the applied imbalanced compensation, the misclassified normal samples disappear. This shows that the proposed AIC model alleviates the problem of normal sample misclassification.

Comparing Figure 7c,d, taking Slider as an example, it is found that, compared with attribute classifier, AIC with the applied imbalanced compensation forms a more compact data distribution. In anomaly detection tasks, the more compact the distribution of normal samples, the lower the density of abnormal samples, which is conducive to abnormal sample detection [32]. This shows that the proposed AIC model helps normal samples form a more compact distribution.

### 4.5. Ensemble

Finally, anomaly detection is performed by fusing the scores of AEIC-MAHA and AIC models through model ensemble. As shown in Table 5, this fusion of generative and discriminative models significantly improves the anomaly detection performance, ranking sixth internationally in the DCASE2023 Challenge Task 2. The score of model fusion can be expressed as
(7)$$S_{ensemble}=S_{AEIC}+\lambda S_{AIC}$$

By changing the value of lambda multiple times, we obtained the optimal performance of the ensemble model. In this work, we chose λ = 0.3. Figure 8 shows the relationship between the value of λ and the AUC of the seven machines. Here, AUC refers to the hmean of AUCs, AUCt, and *p*AUC. Experiments show that the AIC model has high complementarity with the AE model.

## 5. Limitations and Conclusions

Acoustic sensing provides crucial data for ASD in machine condition monitoring. In the MIMII DG [22] dataset, the machines Fan, Valve, Gearbox, Bearing, and Slider were recorded by a TAMAGO-03 microphone. The Fan and Valve were recorded in a sound-proof room, while the Gearbox, Bearing, and Slider were recorded in an anechoic chamber. In the TOYADMOS2 [21] dataset, the ToyCar and ToyTrain were recorded using either a SURE SM11-CN or TOMOCA EM-700 microphone, which introduced some domain shift. Notably, machines other than these seven are unknown, so transferring trained models to unseen machines and enabling anomaly detection is an important area for future research.

While we have utilized all the available datasets from the DCASE Challenge, the experiments are still limited in scope. Regarding audio signals, relatively comprehensive experiments have been conducted to demonstrate the efficacy of the proposed method. However, as the number of samples provided in the datasets is small, it provides challenges to the statistical significance of the results and generalizability of the conclusions. It is noteworthy that simple data augmentation techniques did not lead to significant performance gains for this task. In addition, for other types of signals such as vibration that lack officially provided data, they are not covered in our current work.

Despite the stated limitations, this work makes several valuable contributions. While the datasets are limited, our extensive experiments nonetheless validate the effectiveness of the proposed method for acoustic signals. Our proposed AIC method enables the use of discriminative models for DCASE2023 Task 2, complementing the reconstruction-based AE approach. The AIC model outperforms the baseline AE, presenting an alternative solution direction for this task. Additionally, our proposed IC module is applicable to AE models as well. Incorporating the imbalanced compensation module improves AE-MSE and AE-MAHA to 57.36% and 59.32%, respectively, demonstrating the versatility of our design. Finally, ensembling the AEIC-MAHA and AIC models yields a 9.7% performance gain over the baseline AE-MSE system. This results in a ranking of sixth place internationally in DCASE2023 Challenge Task 2, proving highly competitive. Our contributions not only advance the performance of the algorithm through the novel AIC method but also have a broader impact by enhancing traditional AE models. The consistent improvements across architectures highlight the significance of the ideas introduced in this work.

In conclusion, this work innovatively proposes the AIC framework for acoustic-sensing-based ASD, making it possible to perform anomaly detection with discriminative models without machine ID. The proposed imbalanced compensation module improves both the AIC framework and traditional AE methods significantly. Through integrating AIC and AE using the imbalanced compensation module via model ensemble, we have achieved highly competitive performance. Furthermore, the efficacy of the proposed imbalnce compensation module is further corroborated through t-SNE visualization, which exhibits clear separation between normal and anomalous data. Future work will validate the generalizability of the proposed method on different data types, such as vibration data. Additionally, we will examine the generalization capabilities of our algorithm on other genres of acoustic data.

## Figures and Tables

**Figure 1 sensors-23-08984-f001:**
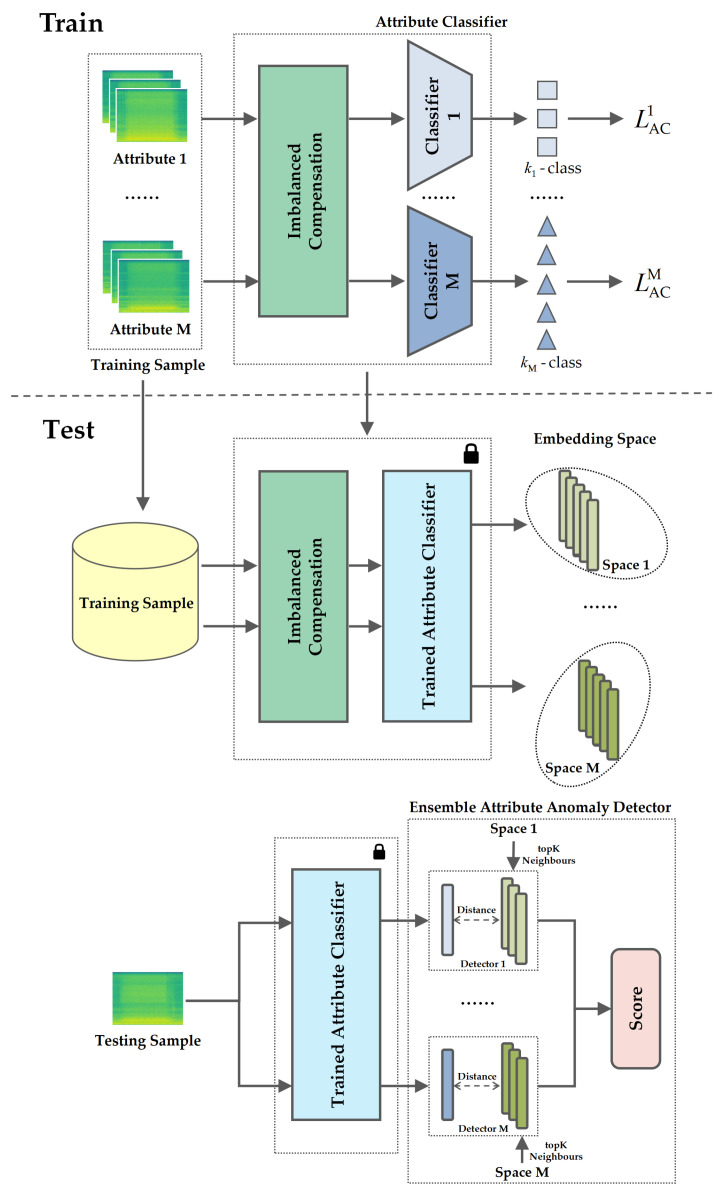
The framework of proposed AIC.

**Figure 2 sensors-23-08984-f002:**
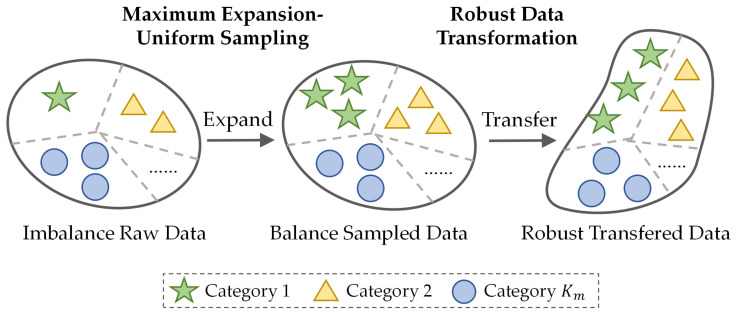
A schematic diagram of the effects of imbalanced compensation on data. ‘Catagory’ means the catagory of operating status.

**Figure 3 sensors-23-08984-f003:**
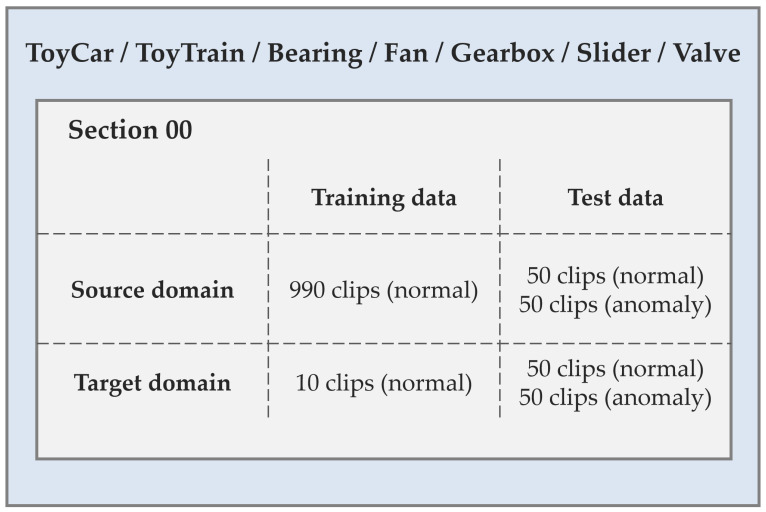
An overview of datasets.

**Figure 4 sensors-23-08984-f004:**
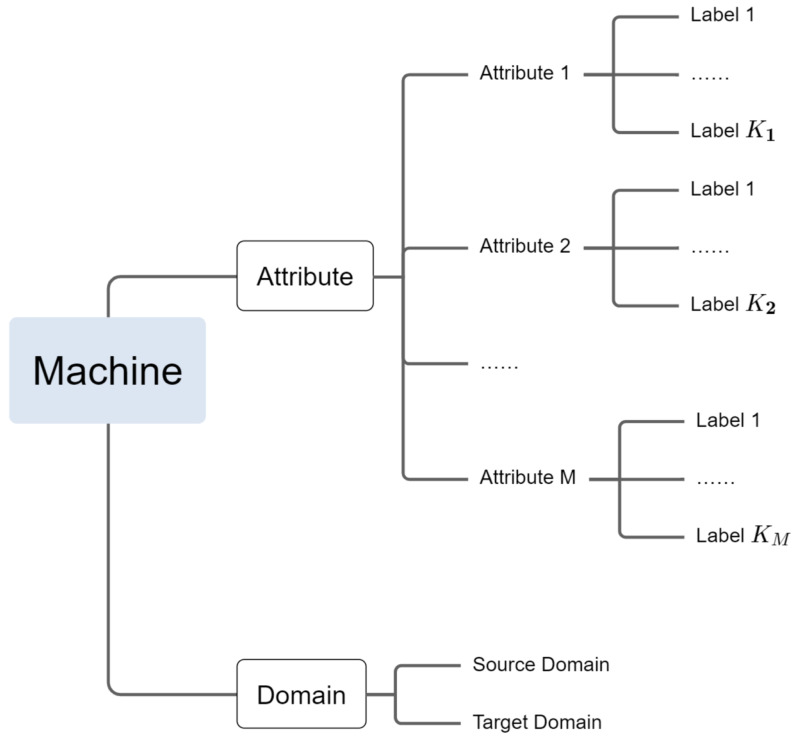
The taxonomy of various labels in the dataset.

**Figure 5 sensors-23-08984-f005:**
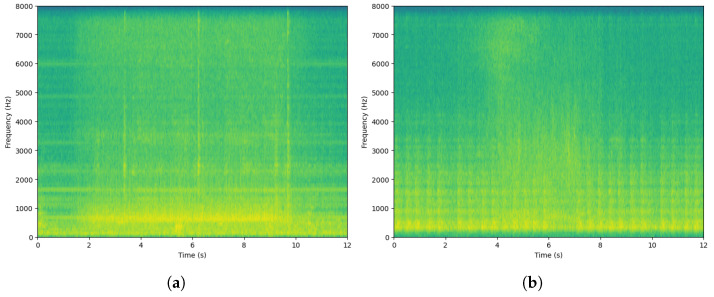
Mel-spectrograms of the 7 machines. (**a**) ToyCar; (**b**) ToyTrain; (**c**) Bearing; (**d**) Fan; (**e**) Gearbox; (**f**) Slider; (**g**) Valve.

**Figure 6 sensors-23-08984-f006:**
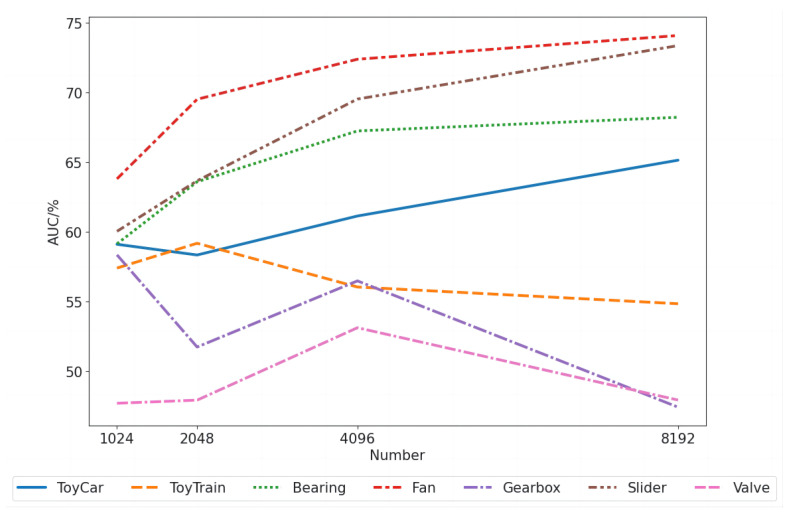
The relationship between the number of samples *R* in the imbalanced compensation module and model performance.

**Figure 7 sensors-23-08984-f007:**
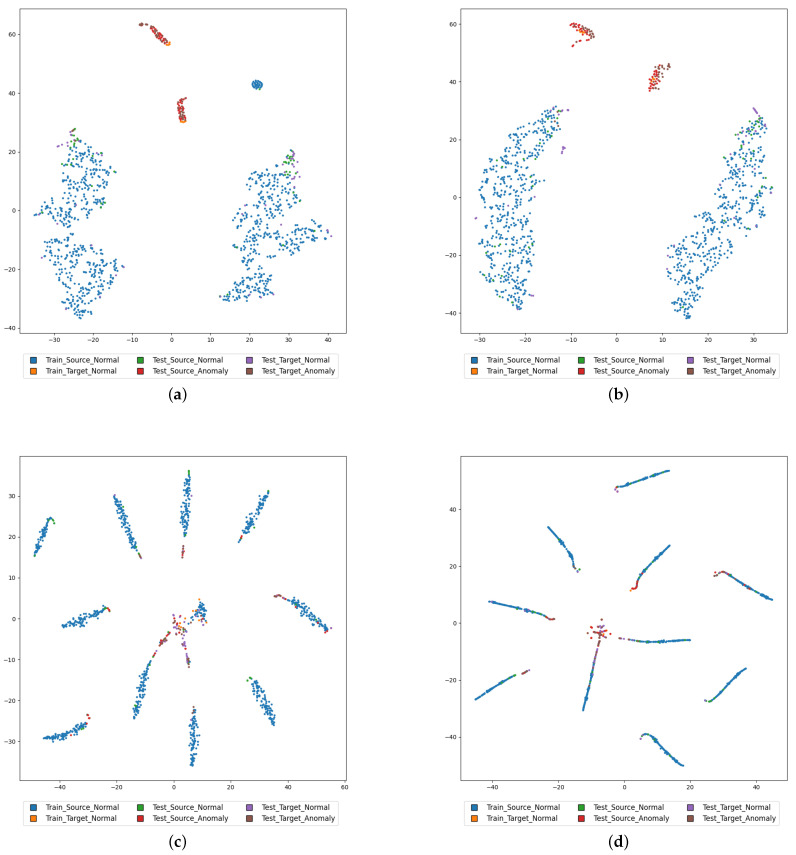
t-SNE visualization comparison between attribute classifier and AIC. (**a**) AC_fan; (**b**) AIC_fan; (**c**) AC_slider; (**d**) AIC_slider.

**Figure 8 sensors-23-08984-f008:**
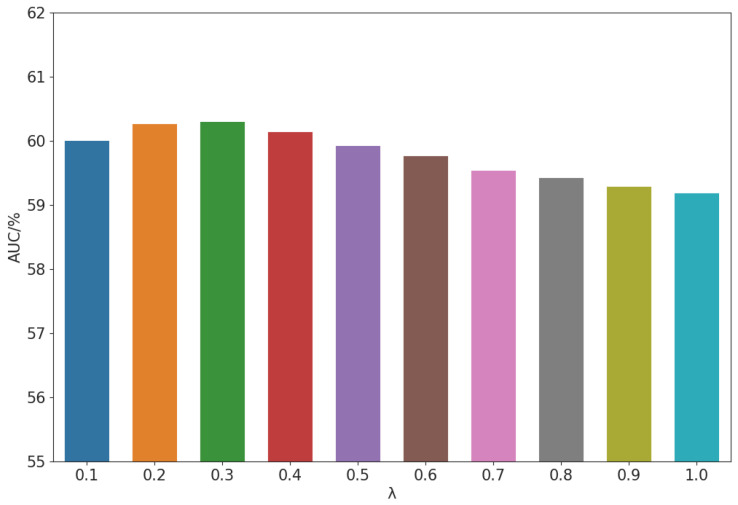
System ensemble performance with the varying of score fusion weight λ.

**Table 1 sensors-23-08984-t001:** Attributes of different machines.

	Attribute 1	Attribute 2	Attribute 3
ToyCar	Car model	Speed	Mic
ToyTrain	Train model	Speed	Mic
Fan	Mixing of machine	N/A	N/A
Gearbox	Voltage	Weight	N/A
Bearing	Velocity	Mic	N/A
Slider	Velocity	Acceleration	N/A
Valve	Open/close	N/A	N/A

**Table 2 sensors-23-08984-t002:** Different operating status labels (‘Category: #samples’) of each attribute in ToyCar dataset.

	Car Model	Speed	Mic
Label 1	C1: 215	3.1 V: 350	1: 990
Label 2	D1: 214	4.0 V: 350	2: 10
Label 3	B1: 166	3.4 V: 290	N/A
Label 4	B2: 164	2.8 V: 5	N/A
Label 5	D2: 116	3.7 V: 5	N/A
Label 6	C2: 115	N/A	N/A
Label 7	A1: 3	N/A	N/A
Label 8	E2: 3	N/A	N/A
Label 9	A2: 2	N/A	N/A
Label 10	E1: 2	N/A	N/A

**Table 3 sensors-23-08984-t003:** Performance of AUC and *p*AUC (*p* = 0.1) comparison on 7 different machines. ‘hmean’ represents the harmonic mean of AUCs, AUCt, and *p*AUC.

	AE-MSE [31]			hmean	AE-MAHA [31]			hmean	AC			hmean	AIC			hmean
	**AUCs**	**AUCt**	p **AUC**		**AUCs**	**AUCt**	p **AUC**		**AUCs**	**AUCt**	p **AUC**		**AUCs**	**AUCt**	p **AUC**	
ToyCar	70.10	46.89	52.47	54.89	74.53	43.42	49.18	52.83	62.37	37.62	50.84	48.17	72.23	50.05	53.89	**57.27**
ToyTrain	57.93	57.02	48.57	54.16	55.98	42.45	48.13	48.23	61.58	61.82	53.78	**58.81**	63.40	48.68	51.42	53.80
Bearing	65.92	55.75	50.42	56.67	65.16	55.28	51.37	56.71	60.98	48.93	52.31	53.62	79.86	54.64	57.47	**62.21**
Fan	80.19	36.18	59.04	52.59	87.10	45.98	59.33	59.90	73.02	28.16	51.26	43.66	65.18	79.62	63.73	**68.82**
Gearbox	60.31	60.69	53.22	57.86	71.88	70.78	54.34	**64.60**	60.55	51.47	54.00	52.40	52.40	60.56	54.15	55.49
Slider	70.31	48.77	56.37	57.18	84.02	73.29	54.72	**68.46**	83.68	58.46	50.78	61.54	82.64	56.44	54.31	62.20
Valve	55.35	50.69	51.18	52.33	56.31	51.40	51.08	**52.83**	69.43	16.34	47.73	31.07	71.38	34.85	49.78	47.78
hmean	64.79	49.59	52.84	55.02	68.84	52.37	52.36	56.91	66.52	35.63	51.45	47.67	68.18	52.12	54.66	**57.53**

**Table 4 sensors-23-08984-t004:** Performance comparisons after applying imbalanced compensation to AE baselines. ‘hmean’ represents the harmonic mean of AUCs, AUCt, and *p*AUC.

	AE-MSE [31]			hmean	AE-MAHA [31]			hmean	AEIC-MSE			hmean	AEIC-MAHA			hmean
	**AUCs**	**AUCt**	p **AUC**		**AUCs**	**AUCt**	p **AUC**		**AUCs**	**AUCt**	p **AUC**		**AUCs**	**AUCt**	p **AUC**	
ToyCar	70.10	46.89	52.47	54.89	74.53	43.42	49.18	52.83	59.64	64.16	51.05	57.76	70.24	59.64	49.21	**58.45**
ToyTrain	57.93	57.02	48.57	**54.16**	55.98	42.45	48.13	48.23	55.48	58.88	48.36	53.87	50.22	48.87	47.73	48.92
Bearing	65.92	55.75	50.42	56.67	65.16	55.28	51.37	56.71	63.98	62.80	51.26	**58.75**	60.38	63.54	52.21	58.31
Fan	80.19	36.18	59.04	52.59	87.10	45.98	59.33	59.90	85.86	62.40	63.78	69.20	81.80	85.44	69.63	**78.35**
Gearbox	60.31	60.69	53.22	57.86	71.88	70.78	54.34	**64.60**	65.64	64.80	54.78	61.32	73.92	69.36	52.10	63.64
Slider	70.31	48.77	56.37	57.18	84.02	73.29	54.72	**68.46**	63.22	47.04	54.94	54.27	79.70	70.61	51.78	65.19
Valve	55.35	50.69	51.18	52.33	56.31	51.40	51.08	**52.83**	51.02	47.98	51.47	50.11	54.40	48.66	51.15	51.30
hmean	64.79	49.59	52.84	55.02	68.84	52.37	52.36	56.91	62.10	57.35	53.30	57.36	65.18	61.51	52.69	**59.32**

**Table 5 sensors-23-08984-t005:** Performance of the system ensemble. ‘hmean’ represents the harmonic mean of AUCs, AUCt, and *p*AUC.

	AEIC-MAHA			hmean	AIC			hmean	Ensemble			hmean
	**AUCs**	**AUCt**	p **AUC**		**AUCs**	**AUCt**	p **AUC**		**AUCs**	**AUCt**	p **AUC**	
ToyCar	70.24	59.64	49.21	58.45	72.23	50.05	53.89	57.27	72.87	57.74	49.78	**58.67**
ToyTrain	50.22	48.87	47.73	48.92	63.40	48.68	51.42	**53.80**	59.22	47.25	48.05	50.97
Bearing	60.38	63.54	52.21	58.31	79.86	54.64	57.47	**62.21**	76.68	58.39	53.10	61.22
Fan	81.80	85.44	69.63	78.35	65.18	79.62	63.73	68.82	80.89	87.16	70.15	**78.76**
Gearbox	73.92	69.36	52.10	63.64	52.40	60.56	54.15	55.49	71.16	69.50	53.94	**63.86**
Slider	79.70	70.61	51.78	65.19	82.64	56.44	54.31	62.20	85.53	69.36	51.94	**66.13**
Valve	54.40	48.66	51.15	51.30	71.38	34.85	49.78	47.78	64.12	42.73	51.42	**51.33**
hmean	65.18	61.51	52.69	59.32	68.18	52.12	54.66	57.53	71.90	58.68	53.34	**60.37**

## Data Availability

The data used in this work were derived from the DCASE2023 Challenge Task 2. All data is available in https://zenodo.org/records/7882613 (accessed on 2 November 2023).

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
