# Peer review of "Acoustic-Sensing-Based Attribute-Driven Imbalanced Compensation for Anomalous Sound Detection without Machine Identity"

_sensors, 2023, doi:10.3390/s23218984_

Round 1

Reviewer 1 Report (Previous Reviewer 2)

Comments and Suggestions for Authors

No comments

Author Response

Thank you for taking the time to thoroughly review our manuscript and for providing valuable feedback. We genuinely appreciate your insights and recognize the areas of improvement you highlighted.

Reviewer 2 Report (New Reviewer)

Comments and Suggestions for Authors

Paper was improved, I see for the yellow, but two things are due to change:

1-      Acoustic sensing provides crucial data for anomalous sound detection (ASD) in condition monitoring is all, so the thing of not knowing the machine must be clarified. The much you know about a source, the better.

2-      Booting ensembles are also an option in many complex problems, see Smart optimization of a friction-drilling process based on boosting ensembles, Journal of Manufacturing Systems 48, 108-121 because you can work with no complete information, please include a discussion and the idea.

3-      The paer format is OK.

Please do the changes, I kindly ask you.

Author Response

Response to Reviewer #2's Comments

Thank you for taking the time to provide your thoughtful feedback on our manuscript. We have fixed all the issues according to your suggestion to improve the quality of revised manuscript. All details are as follows: 

Reviewer's Comment #1:                                  

Acoustic sensing provides crucial data for anomalous sound detection (ASD) in condition monitoring is all, so the thing of not knowing the machine must be clarified. The much you know about a source, the better.

Response:

We thank the reviewer for raising this important point about the need to clarify the machine conditions in our datasets. In the revised manuscript, we have added more details on the recording microphones used for each machine in the MIMII DG and TOYADMOS2 datasets (Line 451-458).

As the reviewer notes, knowledge of the sound source is crucial. To address this, we now explicitly state that the machines outside of the seven mentioned are unknown to us, and propose transfer learning to unseen machines as an important future research direction.

We believe these additions help situate our work and point to next steps in generalized ASD. We appreciate the reviewer highlighting this opportunity to make our machine knowledge assumptions more transparent.

Reviewer's Comment #2:                                  

Booting ensembles are also an option in many complex problems, see Smart optimization of a friction-drilling process based on boosting ensembles, Journal of Manufacturing Systems 48, 108-121 because you can work with no complete information, please include a discussion and the idea.

Response:

We thank the reviewer for the suggestion to discuss model ensemble approaches from the cited paper "Smart optimization of a friction-drilling process based on boosting ensembles." In the revised manuscript, we have added a reference to this work in Line 40-42, and discussed its relevance to our research.

Specifically, we note that model ensemble has proven useful for manufacturing processes under incomplete data conditions. As the reviewer points out, this represents a promising direction for improving ASD robustness. By combining multiple models, the ensemble can overcome limitations of individual models.

We believe discussing the potential of ensemble strategies significantly strengthens our work. We appreciate the reviewer recommending this connection to ensemble methods for ASD with limited operational knowledge.

We genuinely hope these revisions adequately address your concerns, and we believe they substantially improve the overall quality and clarity of the manuscript. Your insights have been crucial in refining our work, and we thank you for your constructive comments and suggestions.

Best regards,

Yifan Zhou, et.al.

Reviewer 3 Report (New Reviewer)

Comments and Suggestions for Authors

This paper proposes Acoustic Sensing-based Attribute-driven Imbalanced Compensation for Anomalous Sound Detection without Machine Identity. In general, this paper is well presented. The following issues can be further considered.

1. More background and motivation of this study can be added, in case the readers are not very familiar with the topic.

2. The descriptions of the well known knowledge can be properly reduced.

3. Why introducing imbalanced compensation based method for the problem? What is the major benefits compared with traditional methods?

4. Some related works on this topic should be reviewed, such as "Robust active learning multiple fault diagnosis of PMSM drives with sensorless control under dynamic operations and imbalanced datasets", "Intelligent Machinery Fault Diagnosis With Event-Based Camera", etc.

5. A couple of ablation studies should be added to evaluate the effects of the key parameters of the proposed method on the performance.

Author Response

Response to Reviewer #3's Comments

First and foremost, we would like to extend our appreciation for the detailed feedback and suggestions offered during the review process. Each point has been carefully considered, and we have made revisions to our manuscript accordingly.

Reviewer's Comment #1:

More background and motivation of this study can be added, in case the readers are not very familiar with the topic.

Response:

We thank the reviewer for the suggestion to expand the background motivation in the introduction. In the revised manuscript, we have added a new paragraph on Lines 17-27 providing more context on the importance of anomalous sound detection and the key challenges it aims to address.

Reviewer's Comment #2:

The descriptions of the well known knowledge can be properly reduced.

Response:

We thank the reviewer for the feedback to reduce repetitious descriptions of well known knowledge. In line with this suggestion, and considering our added introductory background in paragraph 1, we have removed the sentence "The ASD task aims to identify whether the sound emitted from a target machine is normal or anomalous" originally present in paragraph 2 (Line 28). We believe reducing the definition of ASD helps avoid unnecessary repetition.

Reviewer's Comment #3:

Why introducing imbalanced compensation based method for the problem? What is the major benefits compared with traditional methods?

Response:

We thank the reviewer for raising this important question about the motivation behind our proposed imbalanced compensation method. As described in Lines 78-88 and 160-165 of the manuscript, traditional discriminative model based ASD relies on balanced labeled machine ID data, which is unattainable in practice. Our key contribution is enabling discriminative ASD by balancing the imbalanced attribute information that is available. Specifically, our Imbalanced Compensation approach takes highly skewed attribute data and makes it more balanced. This allows discriminative models to be applied for ASD where they previously could not due to imbalanced attribute labels.

Reviewer's Comment #4:

Some related works on this topic should be reviewed, such as "Robust active learning multiple fault diagnosis of PMSM drives with sensorless control under dynamic operations and imbalanced datasets", "Intelligent Machinery Fault Diagnosis With Event-Based Camera", etc.

Response:

We thank the reviewer for recommending the inclusion of additional related works, specifically "Robust active learning multiple fault diagnosis of PMSM drives with sensorless control under dynamic operations and imbalanced datasets" and "Intelligent Machinery Fault Diagnosis With Event-Based Camera."

In accordance with this feedback, we have cited both of these relevant papers on Lines 34-40 of the revised manuscript. Adding discussion of these references significantly expands the breadth of our review of prior work in ASD under imbalanced or limited data.

Reviewer's Comment #5:

A couple of ablation studies should be added to evaluate the effects of the key parameters of the proposed method on the performance.

Response:

We thank the reviewer for the suggestion to add ablation studies evaluating the effects of key parameters in our proposed method. In the original manuscript, we have aimed to provide some analysis of important design choices:

Table 3 shows performance gains after adding the Imbalanced Compensation (IC) module to the Attribute Classifier (AC) module.

Table 4 demonstrates the impact of the IC module specifically on the AE-MSE and AE-MAHA.

Figure 6 illustrates how performance changes with the number of samples used in the IC module.

Figure 8 shows the effect of varying the score fusion weights.

In conclusion, we express our profound gratitude for your insights, which have undoubtedly enriched our manuscript. We are hopeful that our revisions will address your concerns, rendering our paper more comprehensive and worthy of publication.

Best regards,

Yifan Zhou, et.al.

Reviewer 4 Report (New Reviewer)

Comments and Suggestions for Authors

1.      The need of conducting the present study is not clear. Methodology is very complex. There is need to specify how the work presented in the present study is different from the work reported in reference 1 and 2.

2.      Some statements are very confusing, such as “The main challenge of the acoustic sensing-based ASD task is to detect anomalous sounds when only normal sound samples are provided as training data [13], especially for unknown anomalous sounds that did not occur in the given training data.”

3.      “Due to the denoising characteristics of AE itself [7], improving the representation ability of AE may remove anomalies as noise, which limits the anomaly detection performance based on AE”.

The above statement is very suspicious. Rewrite this statement to make it more clear and meaningful.

4.      ’Speed’ has 5 categories from 28V to 40V, what is V with the numerical values?

5.      Cross check the table 1 for attributes of Fan and Gearbox.

6.      Please verify the conclusion drawn from Figure 5, regarding the Stationary signals of Bearing and Fan. Bearing and Fan signals are highly non- stationary and non-linear?

For bearing and gearbox signals you can refer some recent studies, such as:

Integrated approach based on flexible analytical wavelet transform and permutation entropy for fault detection in rotary machines and

A novel feature extraction method based on weighted multi-scale fluctuation based dispersion entropy and its application to the condition monitoring of rotary machines

7.      Why most of the references are of proceedings and conferences? Revise the reference list to include the papers from reputed Journals.

8.      Why some sentences and words are highlighted with yellow color?

Comments on the Quality of English Language

Editing of English language is required,

Author Response

Response to Reviewer #4's Comments

Thank you for taking the time to thoroughly review our manuscript and for providing valuable feedback. We genuinely appreciate your insights and recognize the areas of improvement you highlighted.

Reviewer's Comment #1:

The need of conducting the present study is not clear. Methodology is very complex. There is need to specify how the work presented in the present study is different from the work reported in reference 1 and 2.

Response:

We thank the reviewer for noting the need to clarify the motivation for our study and how it differs from prior works. To address this, in the revised introduction, we have added background on the importance of ASD and the challenges it aims to solve (Lines 17-27). We believe this context helps establish the significance of our research.

To aid readers in understanding how our approach compares to Reference 1 and 2, we now explicitly cite these works when introducing our proposed method (Line 107). Specifically, our study tackles the problem of performing discriminative model based anomaly detection when machine ID labels are unavailable, by utilizing available but imbalanced attribute tags. This enables the use of discriminative models for ASD in situations where they previously could not be applied due to lack of balanced machine IDs.

Reviewer's Comment #2:

Some statements are very confusing, such as “The main challenge of the acoustic sensing-based ASD task is to detect anomalous sounds when only normal sound samples are provided as training data [1–3], especially for unknown anomalous sounds that did not occur in the given training data.”

Response:

We thank the reviewer for pointing out this confusing statement in the original manuscript. In line with the feedback, we have revised the statement on Lines 30-32 as below to make things clear:

The main challenge of the acoustic sensing-based ASD task is to detect anomalous sounds when only normal sound samples are provided as training data [1–3].

Reviewer's Comment #3:

“Due to the denoising characteristics of AE itself [7], improving the representation ability of AE may remove anomalies as noise, which limits the anomaly detection performance based on AE”.

The above statement is very suspicious. Rewrite this statement to make it more clear and meaningful.

Response:

Thanks for pointing out this confusion, we have revised this statement on Lines 57-60 as:

Given the inherent denoising characteristics of AE [12], enhancing the representation capacity of AE may inadvertently treat anomalies as noise, thus constraining the anomaly detection performance reliant on AE.

Reviewer's Comment #4:

’Speed’ has 5 categories from 28V to 40V, what is V with the numerical values?

Response:

We thank the reviewer for catching the lack of explanation around the 'V' units for the speed categories. As noted in reference [21] which introduced this dataset:

"To control operating speed, five voltage levels, 2.8, 3.1, 3.4, 3.7, and 4.0 V, were provided through the stabilized power supply."

Therefore, 'V' refers to voltage levels used to control motor speed. We have added clarification of this on Lines 145-146 in the revised manuscript. Additionally, we apologize for the typo on the voltage values in the original Table 2, and have corrected the decimals places in the updated table. Thank you again for identifying this point needing explanation and correction. It will really help improve understanding and technical accuracy.

Reviewer's Comment #5:

Cross check the table 1 for attributes of Fan and Gearbox.

Response:

We thank the reviewer for encouraging us to double check the attribute definitions for Fan and Gearbox in Table 1. We have cross-validated these descriptions against the DCASE 2023 Challenge documentation https://dcase.community/challenge2023/task-first-shot-unsupervised-anomalous-sound-detection-for-machine-condition-monitoring , which states:

Fan attribute refers to "Mixing of different machine sound between domains."

Gearbox attribute refers to "Different operation voltage and weight attached to the box between domains."

Reviewer's Comment #6:

Please verify the conclusion drawn from Figure 5, regarding the Stationary signals of Bearing and Fan. Bearing and Fan signals are highly non-stationary and non-linear?

For bearing and gearbox signals you can refer some recent studies, such as:

Integrated approach based on flexible analytical wavelet transform and permutation entropy for fault detection in rotary machines and

A novel feature extraction method based on weighted multi-scale fluctuation based dispersion entropy and its application to the condition monitoring of rotary machines.

Response:

We thank the reviewer for the suggested references on analyzing bearing and fan signals, as well as for questioning the stationarity claim about these signals. As recommended, we have read over and cited the suggested works in Lines 43-46.

Regarding stationarity, our initial claim was based on Reference [30]. To address the reviewer's valid skepticism, we re-examined [30] and the official dataset paper [22]. In [30], the bearing and fan signals are indeed processed as stationary. Additionally, inspecting the audio spectrograms in [22], we see reasonable evidence that the signals are stationary, at least over the short segments used.

Reviewer's Comment #7:

Why most of the references are of proceedings and conferences? Revise the reference list to include the papers from reputed Journals.

Response:

We thank the reviewer for the suggestion to include more references from reputable journal publications. You raise a fair point - our original reference list was heavy on conference proceedings papers.

In the fast-moving field of ASD, the latest advancements are often published at venues like ICASSP, along with technical reports from challenges like DCASE. These have timeliness value. However, your feedback has helped us recognize the need to balance these with seminal works from peer-reviewed journals that have undergone rigorous review processes.

In the revised manuscript, we have added citations to the following relevant journal articles:

  • Li, X.; Yu, S.; Lei, Y.; Li, N.; Yang, B. Intelligent Machinery Fault Diagnosis with Event-Based Camera. IEEE Transactions on Industrial Informatics (TII) 2023.
  • Attestog, S.; Senanayaka, J.S.L.; Van Khang, H.; Robbersmyr, K.G. Robust active learning multiple fault diagnosis of PMSM drives with sensorless control under dynamic operations and imbalanced datasets. IEEE Transactions on Industrial Informatics (TII) 2022
  • Bustillo, A.; Urbikain, G.; Perez, J.M.; Pereira, O.M.; de Lacalle, L.N.L. Smart optimization of a friction-drilling process based on boosting ensembles. Journal of Manufacturing Systems (JMS) 2018, 48, 108–121.
  • Sharma, S.; Tiwari, S. A novel feature extraction method based on weighted multi-scale fluctuation based dispersion entropy and its application to the condition monitoring of rotary machines. Mechanical Systems and Signal Processing (MSSPS) 2022, 171, 108909.
  • Sharma, S.; Tiwari, S.; Singh, S. Integrated approach based on flexible analytical wavelet transform and permutation entropy for fault detection in rotary machines. Measurement 2021, 169, 108389.

Reviewer's Comment #8:

Why some sentences and words are highlighted with yellow color?

Response:

We thank the reviewer for the question about the yellow highlighted text in the manuscript. As this submission was a resubmission of a previous version, the highlighting was used to indicate the modifications made since the last review round to facilitate the re-review process. For this revised version, we have again highlighted the edits and new additions in yellow so changes are apparent.

We are genuinely dedicated to enhancing the quality of our work to meet the publication standards and will thoroughly address each point you've raised in our revised submission. Once again, thank you for your constructive critique, which is instrumental in shaping our research to its best possible version.

Best regards,

Yifan Zhou, et.al.

Round 2

Reviewer 3 Report (New Reviewer)

Comments and Suggestions for Authors

Comments are addressed. It can be accepted

Reviewer 4 Report (New Reviewer)

Comments and Suggestions for Authors

All the queries are well addressed. All the best

Comments on the Quality of English Language

No objectionable language errors were observe

This manuscript is a resubmission of an earlier submission. The following is a list of the peer review reports and author responses from that submission.

Round 1

Reviewer 1 Report

Comments and Suggestions for Authors

1. no explained terms AUCs AUCt pAUC hmean, it would be useful to provide them so that the reader does not have to go to other sources. 

No comparisons of the values of all coefficients in the text, only hmean is compared. Please provide reasons why only this coefficient is compared and not the rest. Since they are given in tables it would be useful to refer to them. 

2. Very poor part of the conclusion. It could be combined with the results of the analysis or expanded to make it clearer what was achieved with AIC and whether this kind of reasoning can be transferred to other data sets not only acoustic.

3. not very clear advantage of the AIC model over the rest of the models, from the work it seems that the improvement occurs only for certain subsets of the data - it might be worth checking the performance of the method for other data. e.g. for vibration

Reviewer 2 Report

Comments and Suggestions for Authors

The authors abuses of abbreviations and citations of previous works. The figures do not help to explain anything (figures 1 and 2, for example). The main contributions of the article (Imbalanced Compensation and Ensemble Attribute Anomaly Detector) are very poorly explained. The article must be rewritten in a more intelligible way to be accepted for publication.

Reviewer 3 Report

Comments and Suggestions for Authors

Acoustic sensing provides key data for abnormal sound detection (ASD) in condition monitoring. The discriminative models based on machine identity (ID) classification have shown excellent ASD performance by utilizing strong prior knowledge such as machine ID.Acoustic sensing provides key data for abnormal sound detection (ASD) in condition monitoring. To address this issue, we suggest utilizing unbalanced and inconsistent attribute labels from acoustic sensors, such as machine operating speed and microphone model, serve as weak priors for training attribute classifiers (ACs). Imbalance Compensation (IC) strategy was also introduced to handle extremely imbalanced categories and ensure the trainability of the model. In addition,  a fractional fusion method was proposed to enhance the robustness of anomaly detection.Major revisions required before publication. Some suggestions are as fellow:

1.The text prefix in the block diagram in Figure 1 is reversed.

2.The imbalanced compensation algorithm is not very clear and needs to be supplemented.

3.The number of samples tested in the third part of the experiment is too small, and specific analysis needs to be conducted under specific operating conditions.

Comments on the Quality of English Language

no